# Ultrasound Assessment in Cardiogenic Shock Weaning: A Review of the State of the Art

**DOI:** 10.3390/jcm10215108

**Published:** 2021-10-30

**Authors:** Rebeca Muñoz-Rodríguez, Martín Jesús García-González, Pablo Jorge-Pérez, Marta M. Martín-Cabeza, Maria Manuela Izquierdo-Gómez, Belén Marí-López, María Amelia Duque-González, Antonio Barragán-Acea, Juan Lacalzada-Almeida

**Affiliations:** Cardiology Unit, University Hospital of the Canary Islands, 38320 La Laguna, Spain; martinjgarciagonzalez@gmail.com (M.J.G.-G.); pablorge@gmail.com (P.J.-P.); martamariamc@gmail.com (M.M.M.-C.); mariela_izquierdo@hotmail.com (M.M.I.-G.); marilopezbelen@gmail.com (B.M.-L.); mameliaduque@gmail.com (M.A.D.-G.); barraganacea@gmail.com (A.B.-A.); jlacalzada@gmail.com (J.L.-A.)

**Keywords:** cardiogenic shock weaning, echocardiography, lung ultrasound, diaphragm, ECMO, Impella

## Abstract

Cardiogenic shock (CS) is associated with a high in-hospital mortality despite the achieved advances in diagnosis and management. Invasive mechanical ventilation and circulatory support constitute the highest step in cardiogenic shock therapy. Once established, taking the decision of weaning from such support is challenging. Intensive care unit (ICU) bedside echocardiography provides noninvasive, immediate, and low-cost monitoring of hemodynamic parameters such as cardiac output, filling pressure, structural disease, congestion status, and device functioning. Supplemented by an ultrasound of the lung and diaphragm, it is able to provide valuable information about signs suggesting a weaning failure. The aim of this article was to review the state of the art taking into account current evidence and knowledge on ICU bedside ultrasound for the evaluation of weaning from mechanical ventilation and circulatory support in cardiogenic shock.

## 1. Introduction

Cardiogenic shock (CS) is a complex syndrome defined as a low cardiac output leading to severe end-organ hypoperfusion and progressive failure. Despite the advances in diagnosing, monitoring, and management in the last decade, prognosis remains unacceptably poor with a 35–45% in-hospital mortality [1].

A total of 50–80% of patients with Society for Cardiovascular Angiography and Intervention (SCAI) classification stage C or D cardiogenic shock may require initiation of mechanical ventilation (MV) due to left-ventricular dysfunction and elevated filling pressure leading to pulmonary edema, oxygenation impairment, and the increased work of breathing with ventilatory muscle fatigue [2,3]. The early implantation of mechanical circulatory support devices has increased in recent years as initial vasopressor and inotropic therapies remain insufficient to stabilize the shock status [4]. A total of 19% of the patients with CS following an acute myocardial infarction in the CULPRIT-SHOCK trial received at least one mechanical circulatory support device [5]. Although the exact moment of initiation remains controversial and despite the lack of strong evidence in this field [1,4,6], there are a wide range of hemodynamic, echocardiographic, and clinical parameters that can be assessed to help the heart team make the decision. However, when myocardial contractility improves, the time of weaning from mechanical support, whether ventilatory or circulatory, remains less established, and the weaning criteria and protocols are highly variable across centers [7].

CS weaning includes the withdrawal of both ventilatory and circulatory mechanical support. Nevertheless, their removal does not have to be performed simultaneously. For instance, an awake ECMO strategy, with spontaneous patient breathing, is feasible and safe, being significantly associated with lower MV complication rates and short- and long-term mortality (61.9% vs. 26.7%) [8,9,10].

Echocardiography and right-heart catheterization are the cornerstone for hemodynamic assessment of myocardial improvement [8]. Moreover, lung and diaphragm ultrasound has emerged as a useful tool to predict outcomes of weaning in MV [11]. Although there is a lack of a gold standard [8], a broad range of helpful echocardiographic parameters enable an immediate, low-cost, and noninvasive bedside assessment for the best moment to wean the patient from circulatory and/or ventilatory support. Protocol-guided MV weaning has demonstrated a reduction in reintubation and post-extubation respiratory failure rates [12]. The aim of this study was to provide a review of the current evidence and knowledge on the role of ultrasound when assessing weaning from ventilatory and circulatory mechanical support.

## 2. Ultrasound Assessment in Mechanical Ventilation Weaning

Weaning from MV is challenging in all critically ill patients, even more so when recovering from CS, since concomitant left- and sometimes right-heart failure and diastolic dysfunction are associated with higher rates of extubation failure [13,14]. Heart failure is responsible for 60% of weaning failures [15].

Failure of invasive ventilatory support withdrawal is associated with worse outcomes, independent of the underlying illness severity [16]. Despite clinical tests such as expiratory pressure support ventilation tolerance and spontaneous breathing trials on a T-tube being recommended in weaning guidelines [17], weaning failure rates remain unacceptably high, with around 10–15% of planned extubations failing [13]. Moreover, there is a lack of evidence for this clinical test performed in CS.

### 2.1. Echocardiographic Assessment

Several echocardiographic parameters can be used to predict ventilatory support weaning failure, especially those that allow the estimation of filling pressure and diastolic dysfunction. The influence of the systolic ejection fraction remains unclear, with contradictory results [13,18,19,20].

When assessing diastolic function, obtaining an E/e’ mitral ratio higher than 14.5 is associated with higher rates of weaning failure, even in atrial fibrillation [13,19,20], as are E waves higher than 0.87 m/s [13,21] (Figure 1). However, this method is less reliable in acute decompensated heart failure and left ventricles with larger volumes, where significant mitral regurgitation can lead to underestimation, as well as in resynchronization therapy and wide QRS and the subsequent change in septal e’ due to its abnormal motion [22,23].

The E/A ratio is not useful in critically ill situations, as this parameter frequently suffers from a “pseudonormalization” issue [13], with a difficult quantitative interpretation. However, the presence of a “pseudonormal” or restrictive pattern is related to higher rates of weaning failure [20] (Figure 2). A reduction in the E wave deceleration time below 175 ms, in addition to other parameters which can reflect diastolic impairment, such as raised left-atrial pressure indicated by interatrial septal fixed rightward curvature and left-atrial area larger than 25 cm^2^, is a significant predictor of extubation failure [21]. Moreover, failure is significantly associated with a higher pulmonary capillary edge pressure and elevated pulmonary venous systolic filling [20].

On the other hand, the strain rate and speckle tracking measurements allow to identify impaired systolic dysfunction independent of the loading conditions [24]. Lower left-ventricle strain rate measurements were associated with worse ventilatory weaning outcomes [20]. Nonetheless, its bedside implementation and an expert in echocardiography is not available at all centers.

Mitral regurgitation (MR) has been hypothesized to have a main role in ventilation weaning failure. When there is an underlying functional mechanism of regurgitation, dependent on preload and afterload, it can be undervalued at baseline, while severe MR appears under extubation stress [20]. When functional severe MR is suspected, a stress echocardiography with pharmacological stress can be performed before extubation to rule it out in complex cases [20].

### 2.2. Lung Ultrasound Assessment

The lung ultrasound score (LUS) and modified lung ultrasound score (LUSm) are excellent predictors of weaning failure [11,21,25,26]. They allow a bedside quantification of lung aeration by examining 12 regions for the first or eight regions for the latter [8,27].

Each lung region is given a score; a score of 1 represents a normally aerated region with <3 B-lines, a score of 2 represents a moderately aerated area with ≥3 B-lines, a score of 3 represents severe loss of aeration with multiple B-lines, and a score of 4 represents consolidation (Table 1). On the contrary, the modified version gives a score ranging from 0 if normal to 5 if pleural effusion is observed. The sum of the score given to each region reflects the lung ultrasound score. If the eight-region modified score is performed, a score higher than 7 reflects a higher possibility of weaning failure. In the case of the 12-region score, a score higher than 17 predicts worse extubation outcomes [11,26]. This bedside ultrasound analysis of reverberation artefacts, reflecting impairment of lung aeration secondary to occupied alveolus or sub-optimally recruited, pulmonary congestion or atelectasis have been proposed to be the most powerful predictive factor of extubation failure obtained by ultrasound, being more accurate than the echocardiographic parameters previously explored [25,26,28]. Patients with a high LUS or LUSm score should not be weaned from MV.

No specific score for lung ultrasound assessment has been developed in the context of cardiogenic shock, where pulmonary congestion artefacts could be a limiting issue. However, a combination of lung ultrasound with bedside echocardiography could enhance routine assessment in these complex patients.

### 2.3. Diaphragm Ultrasound

Diaphragm weakness is highly prevalent in long-term mechanically ventilated patients, associated with worse outcomes and difficult weaning [29]. The diaphragm function can be assessed noninvasively by ultrasound, being visualized in the zone of apposition or in a subcostal anterior way [30]. There are two proposed parameters for assessing weaning failure: diaphragmatic excursion (DE), which measures the distance that the diaphragm moves during a spontaneous, unassisted respiratory cycle, and the diaphragm thickening fraction (DTF), which reflects variation in the thickness of the diaphragm as a measure of muscle contraction during the cycle. It is calculated as follows [28,30]:

(thickness at end-inspiration − thickness at end-expiration)/thickness at end-expiration × 100

A reduction of 30% or less in DTF reflects diaphragm fatigue, and it has proven to be a good predictor of worse weaning outcomes. On the other hand, DE is a less accurate parameter, as it can be modified by position, as well as thoracic and abdominal pressure, and it needs to be calculated in the absence of mechanical ventilatory support [26]. Weakness will produce reduced caudal excursion, and paresis will often result in cranial paradoxical excursion during inspiration [30]. There are no homogeneous cutoff values proposed for these parameters, with values ranging from 1 cm to 2.7 cm proposed to predict, in a modest way, weaning failure [26]. As there are no other approachable non-invasive techniques for quantifying the diaphragmatic function in the ICU, echocardiographic assessment constitutes a remarkable skill for the intensivist.

## 3. Ultrasound Assessment in Weaning from Temporary Mechanical Circulatory Support Devices

### 3.1. Veno-Arterial Extracorporeal Membrane Oxygenation (ECMO)

Veno-arterial ECMO support is increasingly used in pharmacological refractory cardiogenic shock as a bridge to recovery, during heart transplantation, or as a bridge to a long-term left-ventricular assist device. There is a current lack of established protocols which can guide or assess the time of withdrawal from mechanical circulatory support during myocardial recovery [31].

Unlike MV weaning assessment, evaluation of systolic function is the key factor in the decision to wean from ECMO. Weaning can be attempted when the ejection fraction of the left ventricle is higher than 35% and/or the left-ventricular outflow tract velocity time integral (VTI) is higher than 15 cm/s, with a minimal ECMO flow under 1.5 L/min or less than 1500 rpm [9]. Previous studies have also proposed lower values of both ejection fraction (around 20–25%) and VTI (10 cm/s) for a successful weaning [32,33] (Figure 3 and Figure 4).

Furthermore, dynamic changes in tissular doppler parameters have been shown to predict successful weaning from ECMO, with an improvement in lateral e′ velocity. These parameters have been proposed as a more accurate predictor of myocardial reserve [32]. Diastolic parameters and the estimation of filling pressures, such as mitral E velocity or its time of deceleration, do not discriminate between successfully weaned patients and failed ones [33].

Assessment of right-ventricle function through the tricuspid annular S′ velocity, the ventricle diameters, and the pulmonary capillary wedge pressure is a strong predictor of outcomes when weaning from veno-arterial ECMO [32,34]. In addition, measuring the right-ventricular and pulmonary circulation coupling by indexing the tricuspid annular S’, TAPSE, and right-ventricle free wall longitudinal strain (RV FLS) to the right-ventricle systolic pressure (RVSP) has recently demonstrated more accuracy in predicting a successful VA-ECMO withdrawal compared to previously described parameters. Values higher than 0.3 of tricuspid annular S’/RVSP and 0,45 of both TAPSE/RVSP and RV FLS/RVSP were significant [35].

In addition, continuous hemodynamic assessment with transesophageal echocardiography, allowing a permanent evaluation of left- and right-ventricle function and volume status, was demonstrated to successfully predict ECMO weaning outcomes [36].

Although there has been an interest in studying the role of LUS role in veno-venous ECMO weaning, especially during the SARS-CoV-2 pandemic, no studies were found wherein lung ultrasound was proposed as a useful tool to predict the success of mechanical circulatory support weaning. The role of the diaphragm and its influence in veno-arterial ECMO weaning have also been studied, whereby a significant relationship was found between the diaphragm thickening fraction and left-ventricle ejection fraction, but without the predictability of successful weaning [37].

### 3.2. Impella

Echocardiographic assessment has a limited role in Impella-assisted patients because of the difficulty in measuring the common Doppler parameters due to the noise generated by the device and its placement at the left-ventricle outflow tract [38]. Nevertheless, readiness to wean can be assessed as a function of some imaging features such as a left-ventricular ejection fraction higher than 25%, an aortic velocity time integral higher than 12 cm/s, or a lateral mitral annulus velocity superior to 6 cm/s [39] (Figure 5).

TEE is commonly used to assist placement, to guide management, and to reveal mechanical complications, as well as to assess the systolic function and concomitant valvulopathies and their severity [40,41].

### 3.3. Intraaortic Balloon Pump

Weaning of an intra-aortic balloon pump (IABP) is usually performed in a hemodynamically assessed fashion by gradually reducing the ratio of augmentation [42]. Although echocardiography can play a role when evaluating improvement of the ejection fraction, cardiac output and filling pressures as well as transesophageal echocardiography are commonly used to guide its placement [43]; however, no specific parameters have demonstrated an accurate predictability when assessing weaning outcomes.

## 4. Conclusions

Prediction of the extubation success can be assessed by bedside echocardiography to estimate diastolic function and filling pressures, suggesting a higher risk of poor outcomes in mechanical ventilatory support withdrawal in cases of an altered E/e’ ratio, mitral E wave, E/A pattern, left-atrial pressure, pulmonary capillary edge pressure, or TDI values. Supplemented with the estimation of the lung ultrasound score and an evaluation of diaphragm weakness, daily, immediate, low-cost, and noninvasive evaluation of ventilatory weaning opportunities can be assessed at the ICU.

Furthermore, when the cardiac index improvement is suspected and weaning from mechanical circulatory support is intended, echocardiography can be a useful tool, especially in ECMO weaning. Improvements of the ejection fraction, VTI, lateral e′ and tricuspid annular S′ velocities, and right-ventricle function are reliable parameters for assessing de-escalation on myocardial support. However, there are no feasible echocardiographic parameters to guide IABP weaning.

## 5. Gaps in Evidence and Research Opportunities

There is a lack of research on echocardiographic alternative parameters for assessing Impella weaning, as well as intra-aortic balloon pump removal, wherein device withdrawal is frequently performed due to failure of the device or a need for mechanical support escalation.

ECMO measurements have been performed in sedated and mechanically ventilated patients, representing a research opportunity for the evaluation of echocardiographic diastolic and systolic parameters and their influence in the weaning of awake ECMO patients.

Lung ultrasound also needs to be potentiated in cardiogenic shock patients. Moreover, there is a need for studies on its role when intending to wean from veno-arterial ECMO.

## Figures and Tables

**Figure 1 jcm-10-05108-f001:**
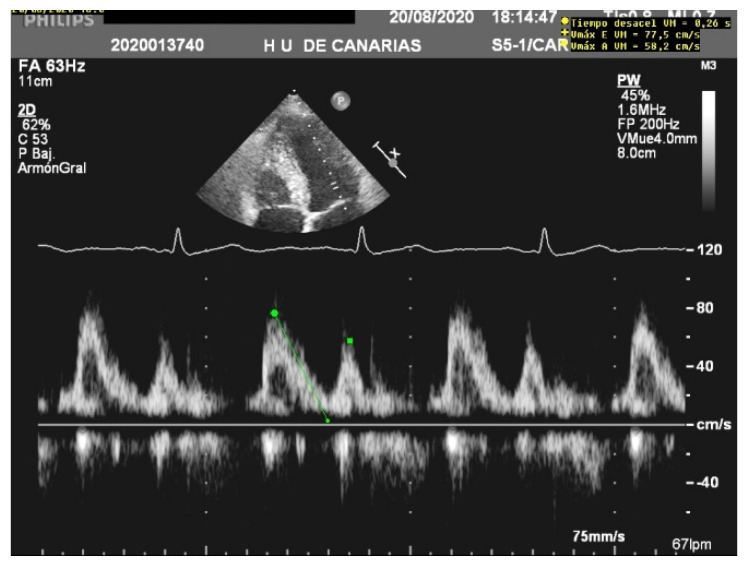
E wave height, deceleration time, and A wave. Normal filling pattern.

**Figure 2 jcm-10-05108-f002:**
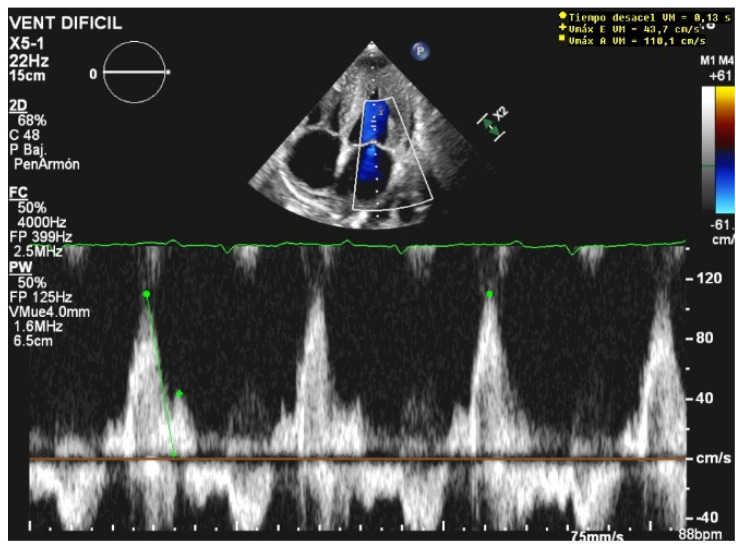
Restrictive diastolic filling pattern.

**Figure 3 jcm-10-05108-f003:**
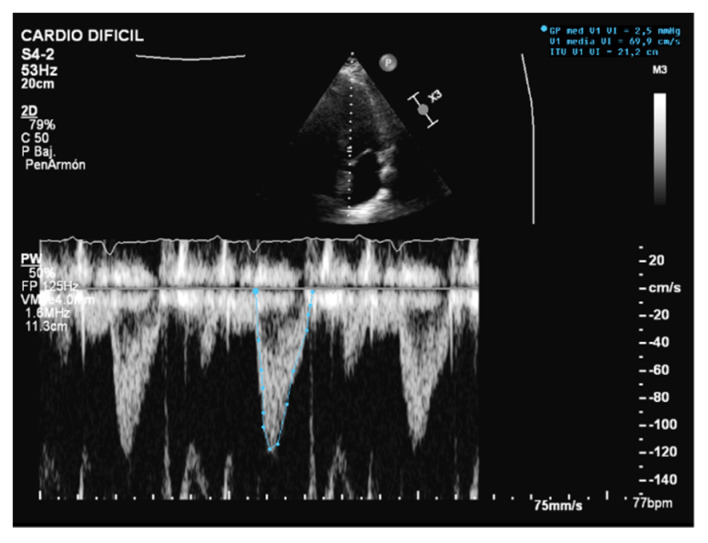
Normal left-ventricular outflow tract velocity integral (VTI).

**Figure 4 jcm-10-05108-f004:**
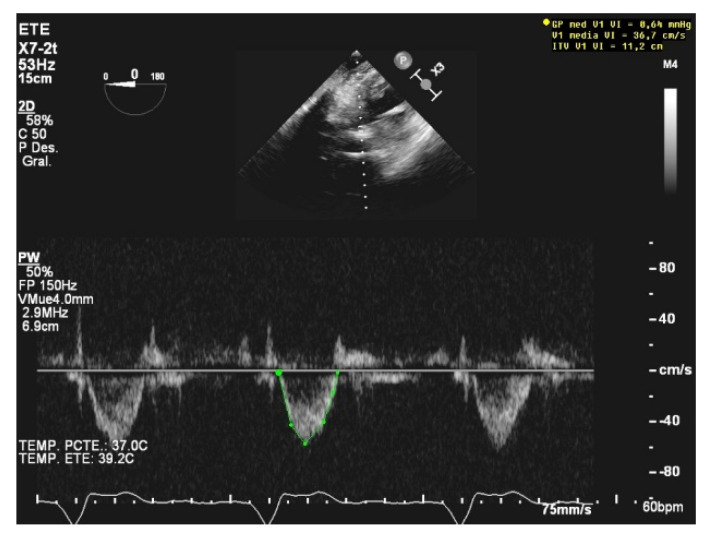
Pathological low left-ventricular outflow tract velocity integral (VTI).

**Figure 5 jcm-10-05108-f005:**
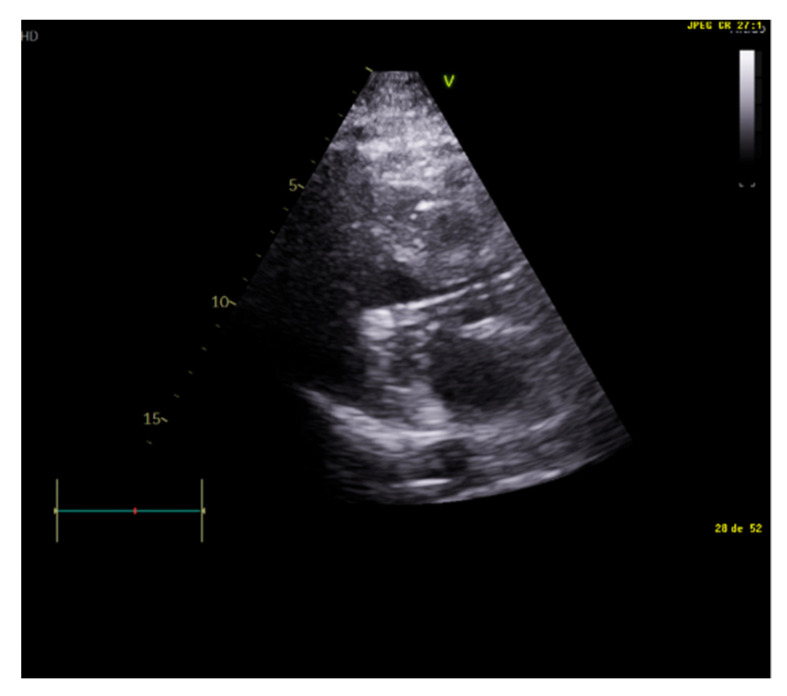
Impella device acoustic noise.

**Table 1 jcm-10-05108-t001:** Lung ultrasound score (LUS).

Points	Ultrasound Pattern	Degree of Aeration
1	<3 B lines	Normally aerated
2	>3 B lines	Moderately aerated
3	Multiple B lines	Severe loss of aeration
4	Consolidation	Complete loss of aeration

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
