# Peer review of "Ultrasound Assessment in Cardiogenic Shock Weaning: A Review of the State of the Art"

_jcm, 2021, doi:10.3390/jcm10215108_

Round 1

Reviewer 1 Report

The authors have chosen an important and timely subject for discussion. The main concern, however, is that the subject is discussed largely disconnected from the underlying physiology, which is at the core of the problem and the driver of clinical decisions and outcomes. This article would be substantially more helpful to the intensivist, if for every echocardiographic measurement the underlying physiological abnormality being assessed be briefly but thoroughly discussed, clearly pointing to reason(s) why the abnormality interferes with the goal of weaning from either mechanical ventilation or circulatory support. In addition, it will be helpful to discuss the usefulness of echocardiography for the titration of therapies which can be, for example, easily perform when the patient is monitor invasively for hemodynamic assessment. In addition, it will be equally important to discuss the operator dependency of these measurements and whether the skills required can be mastered by intensivists to a level that helps achieve the clinical goals stated.

Below, are comments related to specific sections of the manuscript.

Abstract

The following sentence is not clear, please rephrase: “The clinical decision of initiating such support in CS is challenging but more established than the time of weaning”.

Introduction

When discussing the conditions leading to intubation in cardiogenic shock, there is no reference to the increased work of breathing, which beyond oxygenation, is central to the indication for positive pressure ventilation. Increased work of breathing is also central to hemodynamic stability with prior studies showing that increased work of breathing is particularly detrimental under conditions of low blood flow and an indication for initiation of positive pressure ventilation. At the time of attempting liberation from mechanical ventilation, once oxygenation criteria have been met, spontaneous breathing trials assess endurance of respiratory muscles to sustain the work of breathing without additional support, pointing to the importance of discussing the role of respiratory muscles in cardiogenic shock.

Ultrasound Assessment in Mechanical Ventilation Weaning

When referring to the E/é mitral ratio, very specific thresholds are provided. At the same time, it will be helpful to recognize limitations of this index (Park JH, Marwick TH. Use and Limitations of E/e' to Assess Left Ventricular Filling Pressure by Echocardiography. J Cardiovasc Ultrasound. 2011;19:169-73). However, of greater clinical relevance is to elaborate on the physiological measurement that is been assessed by echocardiography and the reasons why such physiological measurement affects weaning from mechanical ventilation. For example, if the problem is the increase in pulmonary capillary hydrostatic pressure, the question becomes in which way it negatively affects weaning. Is it because it’s pulmonary edema that has not been resolved that increases the work of breathing in a situation in which there is decreased diaphragmatic function? In other words, there is a fundamental physiology that drives clinical management and echocardiography represents only a means to identify such physiology, which should not be left unaddressed.

The same concept applies to the discussion on lung ultrasound, what is the fundamental physiological abnormality that ultrasound is detecting and what’s the mechanism by which this abnormality interferes with successful weaning?

It is also worth noticing that the abnormalities described above been detected by echocardiography (e.g., higher left ventricular filling pressures and pulmonary edema/consolidation) can be recognized by many other means that are more familiar to Intensivists. However, assessment of diaphragmatic function represents a challenge without a clear traditional alternative to assess and therefore emphasizing the value that developing such echocardiographic skill would add to the current presentation.

Ultrasound Assessment in Weaning from Temporary Mechanical Circulatory Support Devices

Regarding the left ventricular ejection fraction been a predictor of successful weaning from ECMO, the authors first identify a threshold of 35% but then acknowledged that other authors have identified a lower threshold (i.e., 20 to 25%). What will be worth discussing in this section is that the ejection fraction does not reflect stroke volume and therefore cardiac output. Because the ejection fraction measures only the percentage of blood ejected by the left ventricle during systole, the actual stroke volume is a function of the end-diastolic volume and the anatomic integrity of the mitral valve and aortic valve preventing regurgitation of such stroke volume.

Reviewer 2 Report

I would like to congratulate the authors of this interesting review article. This topic is of importance and enlighted opinions are welcome to help physicians in their daily practice. However, I would have few comment to adress :

  • introduction : 50-80% of CS patients require MV instead of « around 80 » (66% in TRIUMPH; 63% in Hongisto IJC 2017; https://annalsofintensivecare.springeropen.com/articles/10.1186/s13613-019-0571-2)
  • Page 1 :

    I think that the authors should quote the rate (range) of patients treated with mechanical support in contemporary practice

  • Page 1 lower MV complication rate :  lower than ? authors should be more precise

  • Page 3 : 

    "underdiagnosed impaired systolic dysfunction" => 

    CS patients frequently have impaired LV systolic function. This other point making the use of strain less relevant must be cited. 

Page 3 : 

"can be performed before extubation":

The authors should discuss the possibility that patients have chronic left atrial dilatation and the factors that lead to its suspicion

Page 5 : "In addition, measuring the right- 160 ventricular and pulmonary circulation coupling by indexing the tricuspid annular S’, 161 TAPSE, and right-ventricle free wall longitudinal strain to the right-ventricle systolic pres- 162 sure has recently demonstrated more accuracy in predicting a successful VA-ECMO with- 163 drawal compared to previously described parameters" => The threshold values of these different parameters must be quoted in the article.

  • Page 5 : few words on the assessment of the correct placement of Impella would be of interest
  • Page 6 "outcomes". 

    I woud suggest to the authors to add few words on the limited indications of CPIA in contemporary practice. 
